# Event-Triggered Adaptive Fault Tolerant Control for a Class of Uncertain Nonlinear Systems

**DOI:** 10.3390/e22060598

**Published:** 2020-05-27

**Authors:** Chenglong Zhu, Chenxi Li, Xinyi Chen, Kanjian Zhang, Xin Xin, Haikun Wei

**Affiliations:** 1Key Laboratory of Measurement and Control of CSE Ministry of Education, School of Automation, Southeast University, Nanjing 210096, China; chenxi_li@seu.edu.cn (C.L.); chenxinyi@seu.edu.cn (X.C.); xxin@cse.oka-pu.ac.jp (X.X.); hkwei@seu.edu.cn (H.W.); 2Faculty of Computer Science and Systems Engineering, Okayama Prefectural University, 111 Kuboki, Soja, Okayama 719-1197, Japan

**Keywords:** adaptive fault-tolerant control, uncertain strict feedback nonlinear systems, event-triggered control, network data transmission

## Abstract

This paper considers an adaptive fault-tolerant control problem for a class of uncertain strict feedback nonlinear systems, in which the actuator has an unknown drift fault and the loss of effectiveness fault. Based on the event-triggered theory, the adaptive backstepping technique, and Lyapunov theory, a novel fault-tolerant control strategy is presented. It is shown that an appropriate comprise between the control performance and the sensor data real-time transmission consumption is made, and the fault-tolerant tracking control problem of the strict feedback nonlinear system with uncertain and unknown control direction is solved. The adaptive backstepping method is introduced to compensate the actuator faults. Moreover, a new adjustable event-triggered rule is designed to determine the sampling state instants. The overall control strategy guarantees that the output signal tracks the reference signal, and all the signals of the closed-loop systems are convergent. Finally, the fan speed control system is constructed to demonstrate the validity of the proposed strategy and the application of the general systems.

## 1. Introduction

Over the recent decades, as modern technological systems become complex, their corresponding control systems are designed to be more and more sophisticated, including the urgent need to increase the reliability of the systems. This stimulates the study of nonlinear safety-critical control systems. Most research works focus on fault detection and diagnosis (FDD), which can serve as a monitoring system by detecting, localizing, and identifying faults in a system [1]. FDD is a very important procedure, but it is not sufficient to ensure the safe operation of the system. For some safety-critical systems such as aircraft and spacecraft, the continuation of operation is a key feature, and the closed-loop system should be capable of maintaining its pre-specified performance in terms of quality, safety, and stability despite the presence of faults. This calls for the appearance of fault-tolerant control systems (FTCS). More precisely, FTCS is a control system that can accommodate system component faults and is able to maintain system stability and performance in both fault-free and faulty conditions [2,3].Therefore, it has attracted wide attention from scholars and has achieved rapid development; see [4,5] and the references therein. Despite fruitful results for linear systems [6], the fault-tolerant control methods for nonlinear systems have been developed rapidly to compensate the system faults. Viable structure control [7], sliding mode control [8], fuzzy control [9], model predictive control [10], neural networks theory [11], and passivity theory [12] techniques have been proposed for reducing or even eliminating the impact of the failures. From the point of practical application, a significant amount of research on FTC schemes is applied to aerospace aircraft [13], power equipment [14,15], industrial processes [16], etc.

Most of the existing FTC approaches are generally confined to exact modes or uncertain systems with unknown parameters [17]. However, multiple different types of uncertainties and disturbances exist widely in practical nonlinear systems, and treating multiple uncertainties and disturbances as a single equivalent disturbance may result in conservativeness. In this context, the external time-varying disturbances, internal model uncertainties, and unknown control directions are considered in the presence of actuator faults. The following issue is that it is very difficult to compensate uncertainties, disturbances, and faults with a simple controller. Thus, further studies on fault-tolerant control design for uncertain nonlinear systems with disturbance are meaningful and worthwhile. Since the adaptive method was proposed [18,19], it has become an effective method for the uncertainty systems. Due to its favorite advantages such as structural design and strong robustness to uncertainties, adaptive backstepping control originally proposed in [20] has been applied widely to fault-tolerant control for nonlinear systems [21]. For underwater vehicles with thruster fault, the work in [22] developed FTC in the presence of modeling uncertainty and external disturbance. In [23], an adaptive backstepping fault-tolerant control methodology was applied to robot manipulators. The researchers in [24] solved the FTC problem for high-speed trains with unknown system parameters and traction system actuator faults. Although the adaptive backstepping control of uncertain practical systems has received considerable interest, there are still limited works devoted to adaptive fault-tolerant tracking control for more general systems with the external time-varying disturbances, internal model uncertainties, and unknown control directions.

It is worth mentioning that the main limitation of the above-mentioned literature is that the continuous control signals need to be transferred to the actuator in real-time. The traditional periodic sampling control and discrete control can relax the real-time limitation and reduce transmission consumption with the sacrifice of the control performance. However, the unknown actuator faults may result in further poor control performance. Therefore, in order to make the designed controller more practical for real engineering applications, an appropriate compromise between control performance and real-time transmission consumption needs to be considered. In [25], the event-triggered mechanism was proposed to reduce the communication, while still retaining a satisfactory closed-loop behavior, and the results were extended to nonlinear systems in [26]. This type of nonlinear system needs to be converted into the input-to-state stable (ISS) subsystems. In consideration of model uncertainties and unknown external disturbances, the coordinate transform cannot be constructed easily. It should be noted that the ISS assumption is hard to check in practical applications. In [27,28], the fuzzy observer and radial basis function neural network algorithm were explored for a class of uncertain systems in absence of the ISS assumption. Unfortunately, the approximation errors brought about a complex design procedure and the ultimate boundedness of the system rather than asymptotical convergence. An effective event-triggered method was proposed and the global convergence of the deterministic system ensured in [29].

Motivated by the above references, we focus on the event-triggered fault-tolerant control for a general class of feedback systems with the external time-varying disturbances, internal model uncertainties, and unknown control directions. The main contributions of this paper can be sketched as follows. (a) A novel adaptive event-triggered mechanism is investigated to achieve an appropriate compromise between control performance and the real-time transmission consumption. Different from [25,26,29], the unknown drift fault and the loss of effectiveness fault are introduced from the controller to actuator channels. (b) Compared with the above-mentioneded event-triggered methods [27,28], without the ISS assumption, the proposed control schemes in this paper can realize the global convergence instead of only the bound results. Furthermore, a more general and practical system is considered with the external time-varying disturbances, internal model uncertainties, and unknown control directions. This class of systems can be reduced to the system in [29,30].

(c) In this paper, we relax the upper bound information accuracy and measurability limited by the unknown disturbance and faults. The adaptive methods combined with the event-triggered FTC strategy are studied without the traditional function approximations, observer estimation, and the need for excessive control efforts.

The rest of this paper is organized as follows: Section 2 explains the problem formulation in detail and gives some preliminaries. Section 3 presents the main results. Section 4 illustrates the obtained result from practical examples. Section 5 provides some concluding remarks.

## 2. Problem Formulation and Preliminaries

### 2.1. Problem Formulation

The uncertain nonlinear system discussed in this paper is modeled by the following strict feedback form:(1)$$\begin{array}{c} {\dot{x}}_{i}=x_{i+1}+f_{i}\left({\underline{x}}_{i}\right),\;\;\;\;i=1,\cdots ,n-1\\ {\dot{x}}_{n}=af_{n}\left(x\right)+bu_{f}+d_{0}\left(t\right),\\ y=x_{1}, \end{array}$$
where ${\underline{x}}_{i}=:{\left[x_{1},\cdots ,x_{i}\right]}^{T}$, especially ${\underline{x}}_{n}=:{\left[x_{1},\cdots ,x_{n}\right]}^{T}=x\left(t\right)\in {ℜ}^{n}$ is the state of the system, $f_{i}\left(\cdot \right)$ are continuously differentiable system functions, *a* is the unknown system parameter, *b* is the unknown control direction, $d_{0}\left(t\right)$ is the unknown external disturbance, *y* is the system output, and $u_{f}$ is the actuator output. The actuator output $u_{f}$ in a healthy state is equal to the designed control input *u*.

**Remark** **1.**
*If a=b=1 and $d_{0}\left(t\right)=1$ are satisfied, the system (Equation 1) can be simplified to the typical lower triangular structure system without unknown actuator faults described in [29]. In fact, many practical nonlinear systems can be reduced to this kind of structure form [31], such as a jet engine compression system, the electromagnetic floating system, fan speed control system [32], etc. Therefore, the systems we consider in this paper are general and powerful enough to describe many industrial systems.*


When actuator fault occurs, the actuator output $u_{f}$ is no longer equal to the designed control input *u*. The actuator fault can be described as:(2)$$\begin{array}{c} u_{f}\left(t\right)=\lambda \left(t_{\lambda },t\right)u+\mu \left(t_{\mu },t\right) \end{array}$$
where $\lambda \left(t_{\lambda },t\right)\in \left[0,1\right]$ denotes the unknown actuator effectiveness, the same as “healthy indicator” in [33], and $\mu (t_{\mu },t)$ indicates the actuator suddenly bias at the time instant $t_{\mu }$. It is worth noting that the unknown $t_{\lambda },\;t_{\mu },\;\lambda \left(\cdot \right),\;\mu \left(\cdot \right)$ imply fault occurrence instances and that the magnitude of the faults are unmeasurable. In this paper, we consider the condition that 0<λ(·)≤1, i.e., although partially losing its effectiveness, the actuation is still functional such that $u_{f}$ can be influenced by the control input *u* all the time. This is the foundation of the fault-tolerant control system research.

**Remark** **2.**
*Despite there being some existing works on the lower triangular structure systems, for a complicated uncertain system (Equation 1) with external disturbance, how to construct an adaptive fault-tolerant control strategy is still a difficult work. In addition, the restricted network bandwidth limits the direct application of the traditional discrete periodic sampled fault-tolerant control methods. Then, it is a meaningful research challenge to study how to reduce the unnecessary signal in the process of fault-tolerant control and ensure the performance of the system.*


Combining with the actuator fault (Equation 2), the system can be described as the following form:(3)$$\begin{array}{c} {\dot{x}}_{i}=x_{i+1}+f_{i}\left({\underline{x}}_{i}\right),\;\;\;\;i=1,\cdots ,n-1\\ {\dot{x}}_{n}=af_{n}\left(x\right)+b\lambda \left(t_{\lambda },t\right)u+d\left(t\right),\\ y=x_{1}, \end{array}$$
where the integrated uncertainty d(t) is introduced for convenience0.00,0.00,1.00, $d\left(t\right)=d_{0}\left(t\right)+b\mu \left(t_{\mu },t\right)$.

**Remark** **3.**
*The unknown control parameter bλ is very common in dealing with robust control problems [34]. Since the actuator fault parameter λ is brought by equipment aging and mechanical wear in modern engineering applications, the unknown term bλ exhibits the slow time-varying characteristics. A single fixed parameter or time-varying parameter cannot actually describe the fault characteristics of the system. In addition, different from [35], this paper can relax the assumption that the control coefficients are in a known bounded region.*


Control objective: The main purpose of this paper is to design an event-trigger based adaptive fault-tolerant control law for the system (Equation 1) with the uncertain system parameter, unknown disturbance, and unknown control directions, such that the output signal *y* tracks the desired signal $y_{r}$, and all the signals of the system are convergent in spite of the actuator fault (Equation 2).

Two assumptions are described for the system (Equation 3) to design an event-trigger based adaptive fault-tolerant controller:

**Assumption** **1.**
*The nonlinear system function $f_{i}\left(\cdot \right)$ satisfies the Lipschitz condition such that $|f_{i}\left({\underline{x}}_{i}\right)-f_{i}\left({\underline{v}}_{i}\right)|\leq l_{i}∥{\underline{x}}_{i}-{\underline{v}}_{i}∥$, where ${\underline{x}}_{i},\;{\underline{v}}_{i}\in {ℜ}^{i}$, are the state of the system and $l_{i}>0,\;i=1,\cdots n$ are known Lipschitz constants.*


**Assumption** **2.**
*The desired output trajectory signal $y_{r}\left(t\right)$ is continuous, and its $j^{\mathrm{th}}$ order time derivatives $y_{r}^{\left(j\right)}\left(t\right)$ 0.00,0.00,1.00 are assumed to be bounded, that is, $|y_{r}^{\left(j\right)}|\leq k_{rj}$, where $k_{rj}$ are positive constants and j=1,⋯n.*


**Remark** **4.**
*As can be seen from [29] and the reference therein, the locally Lipschitz condition in Assumption 1 has been widely used in tracking control problems and can be satisfied for some practical systems. In addition, this assumption is essential to guarantee the error system is convergent rather than only bounded. Assumption 2 is used commonly to address the reference signal [30,36] and is conventionally true in practice [37]. Contrary to the assumption adopted in [38], Assumption 2 relaxes the boundedness of desired output trajectory signal $y_{r}\left(t\right)$. Hence, Assumptions 1 and 2 are reasonable.*


Before designing the event-triggered controller for the system (Equation 3), motivated by [30], the following adaptive backstepping technique approach is proposed to design the fault-tolerant controller:

### 2.2. Preliminaries

The following classical coordinate transformation is introduced:(4)$$\begin{array}{c} z_{1}=x_{1}-y_{r} \end{array}$$(5)$$\begin{array}{c} z_{i}=x_{i}-y_{r}^{\left(i-1\right)}-\alpha_{i-1},\;\;\;\;i=2,\cdots n \end{array}$$
where $\alpha_{i-1}$ are virtual control, $x_{i}$ and $z_{i}$ are the corresponding error variables, and $y_{r}^{\left(i\right)}$ denote the $i^{\mathrm{th}}$ time derivatives of the variable $y_{r}$.

Step 1. Taking the time derivative for the tracking error $z_{1}$:(6)$$\begin{array}{c} {\dot{z}}_{1}={\dot{x}}_{1}-{\dot{y}}_{r}=x_{2}-y_{r}^{\left(1\right)}+f_{1}=z_{2}+f_{1}+\alpha_{1} \end{array}$$

We can design the virtual control law $\alpha_{1}$ as:(7)$$\begin{array}{c} \alpha_{1}=-c_{1}z_{1}-f_{1} \end{array}$$
where $c_{1}$ is a positive design parameter. Then, we obtain:(8)$$\begin{array}{c} z_{1}{\dot{z}}_{1}=-c_{1}z_{1}^{2}+z_{1}z_{2}. \end{array}$$

Step i. (i=2,⋯n−1)
(9)$$\begin{array}{c} {\dot{z}}_{i}=x_{i+1}-y_{r}^{\left(i\right)}+f_{i}-{\dot{\alpha }}_{i-1}=z_{i+1}+f_{i}+\alpha_{i}-{\dot{\alpha }}_{i-1} \end{array}$$

We design the virtual control law $\alpha_{i}$ as:(10)$$\begin{array}{c} \alpha_{i}=-c_{i}z_{i}-f_{i}-z_{i-1}+{\dot{\alpha }}_{i-1} \end{array}$$
where $c_{i}$ is a positive design parameter. Then,
(11)$$\begin{array}{c} z_{i}{\dot{z}}_{i}=-c_{1}z_{i}^{2}+z_{i}z_{i+1}-z_{i-1}z_{i} \end{array}$$

Step n.
(12)$$\begin{array}{c} {\dot{z}}_{n}=\lambda bu-y_{r}^{\left(n\right)}+af_{n}+d-{\dot{\alpha }}_{n-1} \end{array}$$

We design the control law *u* as:(13)$$\begin{array}{c} \lambda bu=-c_{n}z_{n}-af_{n}-z_{n-1}+{\dot{\alpha }}_{n-1}+y_{r}^{\left(n\right)}-d \end{array}$$
where $c_{n}$ is a positive design parameter. Then, we obtain:(14)$$\begin{array}{c} z_{n}{\dot{z}}_{n}=-c_{n}z_{n}^{2}-z_{n-1}z_{n} \end{array}$$

The unknown control coefficient can be redefined as λb=1/h. Then, the adaptive fault-tolerant control law is designed as follows: (15)$$\begin{array}{cc} & u=\hat{h}\bar{u} \end{array}$$(16)$$\begin{array}{c} \bar{u}=-c_{n}z_{n}-\hat{a}f_{n}-z_{n-1}+{\dot{\alpha }}_{n-1}+y_{r}^{\left(n\right)}-sgn\left(z_{n}\right)\hat{D} \end{array}$$
where *D* is the unknown upper bound of d(t) and $\hat{D},\;\hat{a},\;\hat{h}$ is the estimate of unknown system parameters D,a,h. Under the control law and redefining the estimate error as $\tilde{a}=a-\hat{a},\;\tilde{b}=b-\hat{b},\;\tilde{h}=h-\hat{h}$, we have:(17)$$\begin{array}{c} z_{n}{\dot{z}}_{n}=-c_{n}z_{n}^{2}-z_{n-1}z_{n}+z_{n}\tilde{a}f_{n}-z_{n}\lambda b\tilde{h}\bar{u}+z_{n}d-z_{n}sgn\left(z_{n}\right)\hat{D} \end{array}$$

We take the Lyapunov candidate function as:(18)$$\begin{array}{c} V=\left(\sum_{i=1}^{n}\frac{1}{2}z_{i}^{2}\right)+\frac{1}{2\beta }{\tilde{a}}^{2}+\frac{\lambda b}{2\gamma }{\tilde{h}}^{2}+\frac{1}{2\delta }{\tilde{D}}^{2} \end{array}$$

Then, the time derivative of *V* along with the system and the control law is calculated by:(19)$$\begin{array}{ccc} \dot{V} & = & \sum_{i=1}^{n}z_{i}{\dot{z}}_{i}+\frac{1}{\beta }\tilde{a}\dot{\tilde{a}}+\frac{\lambda b}{\gamma }\tilde{h}\dot{\tilde{h}}+\frac{1}{\delta }\tilde{D}\dot{\tilde{D}}\\ & = & -\sum_{i=1}^{n}c_{i}z_{i}^{2}+\frac{1}{\beta }\tilde{a}\left(\beta z_{n}f_{n}-\dot{\hat{a}}\right)+\frac{\lambda b}{\gamma }\tilde{h}\left(-\gamma z_{n}\bar{u}-\dot{\hat{h}}\right)+z_{n}d-z_{n}sgn\left(z_{n}\right)\left(D-\tilde{D}\right)-\frac{1}{\delta }\tilde{D}\dot{\hat{D}} \end{array}$$

The adaptive update laws are designed as follows: (20)$$\begin{array}{ccc} \dot{\hat{a}} & = & \beta z_{n}f_{n} \end{array}$$(21)$$\begin{array}{ccc} \dot{\hat{h}} & = & -\gamma z_{n}\bar{u} \end{array}$$(22)$$\begin{array}{ccc} \dot{\hat{D}} & = & \delta z_{n}sgn\left(z_{n}\right) \end{array}$$

**Lemma** **1.**
*Considering the uncertain nonlinear system (Equation 1) with external disturbance, the actuator fault (Equation 2), and unknown control directions, Assumption 2 is satisfied. The proposed adaptive fault-tolerant controller (Equation 15) and (16) and the parameter update laws (Equation 20)–(22) can ensure that all the signals in the closed-loop system are bounded and the asymptotic tracking is achieved, i.e.,*
(23)$$\begin{array}{c} \lim_{t\to \infty }\left[y\left(t\right)-y_{r}\left(t\right)\right]=0. \end{array}$$


**Proof.** Substituting the adaptive laws (Equation 20)–(Equation 22) into (Equation 20) yields:
(24)$$\begin{array}{ccc} \dot{V} & = & -\sum_{i=1}^{n}c_{i}z_{i}^{2}+z_{n}d-z_{n}sgn\left(z_{n}\right)D\\ & \leq & -\sum_{i=1}^{n}c_{i}z_{i}^{2} \end{array}$$It is obvious that *V* is non-increasing. Together with (Equation 18), the closed-loop system signal $z_{i}\;$ is bounded. By invoking the LaSalle invariance principle, we can conclude that $z_{i}\left(t\right)\to 0,\;i=1,\cdots n$ as t→∞. This shows that the asymptotic tracking is achieved. ☐

## 3. The Main Results

The basic architecture for event-triggered adaptive fault-tolerant control is shown in Figure 1. It is obvious that the introduced event-trigger control mechanism is different from the traditional fault-tolerant control system. Whether the real-time data of the sensors can be transmitted is determined by the event-triggered mechanism.

**Remark** **5.**
*Compared with the traditional time-triggered signal transmission strategy [39,40], the event-triggered mechanism can reduce the frequency of sending redundant control signals in the system, restrain unnecessary information transmission, and make full use of bandwidth resources. The collaborative design of the event trigger and the controller can effectively ensure the fault tolerance performance of the system. In addition, the adaptive update law can relax the parameters and disturbance assumption and enhance the practicability of the method in this paper.*


Under the event-triggered sample, the state vector can be written as:(25)$$\begin{array}{ccc} {\hat{x}}_{i}\left(t^{+}\right) & = & x_{i}\left(t_{k}\right),\;\;\;\;t=t_{k}. \end{array}$$(26)$$\begin{array}{ccc} {\hat{x}}_{i}\left(t\right) & = & x_{i}\left(t_{k}\right),\;\;\;\;t\in \left[t_{k},\;t_{k+1}\right),\;\;i=1,\cdots ,n \end{array}$$

The event is triggered at $t_{k}$, and the status data received by the controller are kept as constant by the zero-order hold at the time interval $\left[t_{k},\;t_{k+1}\right)$. Then, the event-triggering conditions are designed as: (27)$$\begin{array}{ccc} ∥{\tilde{x}}_{1}∥ & \leq & \frac{c_{1}}{c_{1}+l_{1}}\zeta ∥z_{1}∥ \end{array}$$(28)$$\begin{array}{ccc} ∥{\tilde{\underline{x}}}_{i}∥ & \leq & \frac{c_{i}}{1+c_{i}+l_{i}}\zeta ∥z_{i}∥,\;\;i=2,\cdots ,n-1 \end{array}$$(29)$$\begin{array}{ccc} ∥{\tilde{\underline{x}}}_{n}∥ & \leq & \frac{c_{n}}{1+c_{n}+\bar{a}l_{n}}\zeta ∥z_{n}∥ \end{array}$$
where the parameter ζ is an adjustable positive constant to be specified later, $c_{i}$ are the positive virtual control parameters in (Equation 7), (Equation 10) and (Equation 12), $\bar{a}$ is the estimated upper bound of parameter *a*, and $l_{i}$ are the Lipschitz constants in Assumption 1. When the event-triggering conditions (Equation 27)–(Equation 29) are violated, the control signal $u(t_{k})$ will be updated to $u(t_{k+1})$, and it is maintained until the next event triggering at $t_{k+2}$. Then, the event-triggered errors,
(30)$$\begin{array}{c} {\tilde{x}}_{i}\left(t\right)=x_{i}\left(t\right)-{\hat{x}}_{i}\left(t\right),\;\;\;\;t\in \left[t_{k},\;t_{k+1}\right),\;\;i=1,\cdots ,n \end{array}$$

The event-triggered based control variables are as follows:(31)$$\begin{array}{c} {\hat{z}}_{1}\left(t\right)={\hat{x}}_{1}-y_{r}, \end{array}$$(32)$$\begin{array}{c} {\hat{z}}_{i}\left(t\right)={\hat{x}}_{i}-y_{r}^{\left(i-1\right)}-{\hat{\alpha }}_{i},\;\;i=2,\cdots ,n-1. \end{array}$$
where the virtual control ${\hat{\alpha }}_{i}$ can be redesigned later. Then, the event-triggered adaptive control law is designed as follows:(33)$$\left\{\begin{array}{c} u=\hat{h}\bar{u}\;\;\;\;\;\;\;\;\;\;\;\;\;\;\;\;\;\;\;\;\;\;\;\;\;\;\;\;\;\;\;\;\;\;\;\;\;\;\;\;\;\;\;\;\;\;\;\;\;\;\;\;\;\;\;\;\;\;\;\;\;\;\;\;\;\;\;\;\;\;\;\\ \bar{u}=-c_{n}{\hat{z}}_{n}-\hat{a}f_{n}\left({\hat{\underline{x}}}_{n}\right)-{\hat{z}}_{n-1}+y_{r}^{\left(n\right)}-sgn\left(z_{n}\right)\hat{D} \end{array}\right.$$
where the parameter update laws are the same as (Equation 20)–(Equation 22).

The event-triggered virtual variable errors:(34)$$\begin{array}{ccc} {\tilde{z}}_{1} & = & z_{1}-{\hat{z}}_{1}=x_{1}-y_{r}-{\hat{x}}_{1}+y_{r}=x_{1}-{\hat{x}}_{1}={\tilde{x}}_{1} \end{array}$$(35)$$\begin{array}{ccc} {\tilde{z}}_{2} & = & z_{2}-{\hat{z}}_{2}=x_{2}-y_{r}^{\left(1\right)}-{\hat{\alpha }}_{1}-{\hat{x}}_{2}+y_{r}^{\left(1\right)}+{\hat{\alpha }}_{1}=x_{2}-{\hat{x}}_{2}={\tilde{x}}_{2} \end{array}$$

It is easy to conclude that:(36)$$\begin{array}{ccc} {\tilde{z}}_{i} & = & {\tilde{x}}_{i}\;\;\;\;i=1,2\cdots ,n. \end{array}$$

The main result of in this paper is given in the following theorem:

**Theorem** **1.**
*For the uncertain nonlinear system (Equation 1) with unknown external disturbance and actuator faults (Equation 2), if the system satisfies Assumption 1 and the reference signal satisfies Assumption 2, the events are triggered when the conditions (Equation 27)–(Equation 29) are violated under the proposed adaptive fault-tolerant controller (Equation 33), while the parameter update laws (Equation 20)–(Equation 22) can ensure that all the signals in the closed-loop system are bounded, and the asymptotic tracking is achieved.*


**Proof.** Following Lemma 1, we complete the proof of Theorem 1 by treating two cases. In Case 1, we focus on the inter-event intervals. In Case 2, the system is analyzed at the event-triggered instants.Case 1: Considering the inter-event intervals $t_{k}\leq t<t_{k+1},k\in Z^{+}$, the events for which sensor data are transmitted to the controller via the network are not enough to be triggered under the event conditions (Equation 27)–(Equation 29). Recalling the continuous adaptive fault-tolerant proof procedure, we demonstrate the proposed adaptive fault-tolerant controller $u\left(t\right)=u\left(t_{k}\right)$ to guarantee the error signal convergence.Step 1. In this step, following the traditional virtual control $\alpha_{1}$, the virtual control ${\hat{\alpha }}_{1}$ is redesigned to compensate the event-triggered error under the event condition (Equation 27).We take the Lyapunov candidate function as $V_{1}=\frac{1}{2}z_{1}^{2}+\frac{1}{2}{\hat{z}}_{1}^{2}$ and the virtual control ${\hat{\alpha }}_{1}=-c_{1}{\hat{z}}_{1}-f_{1}\left({\hat{x}}_{1}\right)-k_{1}sgn\left(z_{1}\right)$. Calculating the time derivative of $V_{1}$ along solutions of System (1), it follows from Assumption 1 that:
(37)$$\begin{array}{ccc} {\dot{V}}_{1} & = & z_{1}\left(z_{2}+f_{1}+{\hat{\alpha }}_{1}\right)-\left(z_{1}-{\tilde{z}}_{1}\right){\dot{y}}_{r}\\ \; & = & z_{1}z_{2}+z_{1}\left(f_{1}-c_{1}{\hat{z}}_{1}-f_{1}\left({\hat{x}}_{1}\right)-k_{1}sgn\left(z_{1}\right)\right)+\left({\tilde{x}}_{1}-z_{1}\right){\dot{y}}_{r}\\ \; & = & z_{1}z_{2}+z_{1}\left(-c_{1}z_{1}+c_{1}\left(z_{1}-{\hat{z}}_{1}\right)+f_{1}\left(x_{1}\right)-f_{1}\left({\hat{x}}_{1}\right)\right)+\left({\tilde{x}}_{1}-z_{1}\right){\dot{y}}_{r}-k_{1}\left|z_{1}\right|\\ \; & \leq & z_{1}z_{2}-c_{1}z_{1}^{2}+|z_{1}|\left(c_{1}+l_{1}\right)|{\tilde{x}}_{1}\left|+\right|{\tilde{x}}_{1}\left|\right|{\dot{y}}_{r}|-(k_{1}-|{\dot{y}}_{r}\left|)\right|z_{1}| \end{array}$$If we choose the event-triggered conditions as (Equation 27), then (Equation 37) can be calculated as:
(38)$$\begin{array}{ccc} {\dot{V}}_{1} & \leq & z_{1}z_{2}-c_{1}z_{1}^{2}+c_{1}\zeta z_{1}^{2}-(k_{1}-|{\dot{y}}_{r}|+\frac{c_{1}}{c_{1}+l_{1}}\zeta |{\dot{y}}_{r}\left|)\right|z_{1}| \end{array}$$Combining with Assumption 2 the boundedness of ${\dot{y}}_{r}$, we can design $k_{1}$ with $k_{1}-|{\dot{y}}_{r}|+\frac{c_{1}}{c_{1}+l_{1}}\zeta \left|{\dot{y}}_{r}\right|>0$, so that:
(39)$$\begin{array}{ccc} {\dot{V}}_{1} & \leq & -\left(1-\zeta \right)z_{1}^{2}+z_{1}z_{2} \end{array}$$This completes the proof of Step 1.Step 2. Different from Step 1, a virtual control ${\hat{\alpha }}_{2}$ that meets the requirements is designed based on the triggering conditions (Equation 29).We first choose the virtual control ${\hat{\alpha }}_{2}=-c_{2}{\hat{z}}_{2}-f_{2}\left({\hat{\underline{x}}}_{2}\right)-{\hat{z}}_{1}-k_{2}sgn\left(z_{2}\right)$. Calculating the time derivative of the Lyapunov candidate function $V_{2}=V_{1}+\frac{1}{2}z_{2}^{2}+\frac{1}{2}{\hat{z}}_{2}^{2}$ along the solutions of System (Equation 3), it follows from Assumptions 1 and 2 that:
(40)$$\begin{array}{ccc} {\dot{V}}_{2} & = & {\dot{V}}_{1}+z_{2}{\dot{z}}_{2}+{\hat{z}}_{2}{\dot{\hat{z}}}_{2}=z_{2}\left(z_{3}+f_{2}+\alpha_{2}\right)-\left(z_{2}-{\tilde{z}}_{2}\right)y_{r}^{\left(2\right)}\\ \; & = & {\dot{V}}_{1}+z_{2}z_{3}+z_{2}\left(f_{2}-{\hat{z}}_{1}-c_{2}{\hat{z}}_{2}-f_{2}\left(\underline{{\hat{x}}_{2}}\right)-k_{2}sgn\left(z_{2}\right)\right)+\left({\tilde{x}}_{2}-z_{2}\right)y_{r}^{\left(2\right)}\\ \; & \leq & -\left(1-\zeta \right)c_{1}z_{1}^{2}+z_{2}z_{3}+z_{2}(\left(z_{1}-{\hat{z}}_{1}\right)-c_{2}z_{2}+c_{2}\left(z_{2}-{\hat{z}}_{2}\right)\\ \; & \; & +f_{2}\left(\underline{x_{2}}\right)-f_{2}\left(\underline{{\hat{x}}_{2}}\right))+\left({\tilde{x}}_{2}-z_{2}\right)y_{r}^{\left(2\right)}-k_{2}\left|z_{2}\right|\\ & \leq & z_{2}z_{3}-\left(1-\zeta \right)c_{1}z_{1}^{2}-c_{2}z_{2}^{2}+z_{2}\left({\tilde{x}}_{1}+c_{2}{\tilde{x}}_{2}+L_{2}\underline{{\tilde{x}}_{2}}\right)+\left({\tilde{x}}_{2}-z_{2}\right)y_{r}^{\left(2\right)}-k_{2}\left|z_{2}\right|\\ \; & \leq & z_{2}z_{3}-\left(1-\zeta \right)c_{1}z_{1}^{2}-c_{2}z_{2}^{2}+|z_{2}|\left(1+c_{2}+L_{2}\right)\left|\right|\underline{{\tilde{x}}_{2}}\left|\right|+\left|\right|\underline{{\tilde{x}}_{2}}\left||\right|y_{r}^{\left(2\right)}|-(k_{2}-|y_{r}^{\left(2\right)}\left|)\right|z_{2}| \end{array}$$If the event-triggered condition is designed as
(41)$$\begin{array}{c} {\dot{V}}_{2}\leq z_{2}z_{3}-\left(1-\zeta \right)c_{1}z_{1}^{2}-c_{2}z_{2}^{2}+c_{2}\zeta z_{2}^{2}-(k_{2}-|y_{r}^{\left(2\right)}|+\frac{c_{2}}{1+c_{2}+l_{2}}\zeta |y_{r}^{\left(2\right)}\left|)\right|z_{2}| \end{array}$$Together with Assumption 2, that the boundedness of $y_{r}^{\left(i\right)},\;\;i=1,\cdots n-1$, we can design $k_{2}$ with $k_{2}-|y_{r}^{\left(2\right)}|+\frac{c_{2}}{1+c_{2}+l_{2}}\zeta \left|y_{r}^{\left(2\right)}\right|>0$, hence,
(42)$$\begin{array}{ccc} {\dot{V}}_{2} & \leq & -\left(1-\zeta \right)\sum_{i=1}^{2}z_{i}^{2}+z_{2}z_{3} \end{array}$$Step n–1. In this step, the mathematical induction is introduced to design virtual control ${\hat{\alpha }}_{n-1}$ under the trigger conditions (29).Suppose at Step n−2 that there exists a suitable threshold strategy to satisfy:
(43)$$\begin{array}{ccc} {\dot{V}}_{n-2} & \leq & -\left(1-\zeta \right)\sum_{i=1}^{n-2}z_{i}^{2}+z_{n-2}z_{n-1} \end{array}$$Then, we take the virtual control:
(44)$$\begin{array}{c} {\hat{\alpha }}_{n-1}=-c_{n-1}{\hat{z}}_{n-1}-f_{n-1}\left({\hat{\underline{x}}}_{n-1}\right)-{\hat{z}}_{n-2}-k_{n-1}sgn\left(z_{n-1}\right). \end{array}$$Calculating the time derivative of Lyapunov candidate function $V_{n-1}=V_{n-2}+\frac{1}{2}z_{n-1}^{2}+\frac{1}{2}{\hat{z}}_{n-1}^{2}$ along solutions of System (1), it follows from Assumptions 1 and 2 that:
(45)$$\begin{array}{ccc} {\dot{V}}_{n-1} & = & {\dot{V}}_{n-2}+z_{n-1}{\dot{z}}_{n-1}+{\hat{z}}_{n-1}{\dot{\hat{z}}}_{n-1}\\ \; & = & {\dot{V}}_{n-2}+z_{n-1}\left(z_{n}+f_{n-1}+\alpha_{n-1}\right)-\left(z_{n-1}-{\tilde{z}}_{n-1}\right)y_{r}^{\left(n-1\right)}\\ \; & = & {\dot{V}}_{n-2}+z_{n-1}z_{n}+z_{n-1}(f_{n-1}-{\hat{z}}_{n-2}-c_{n-1}{\hat{z}}_{n-1}\\ \; & & -f_{n-1}\left({\underline{\hat{x}}}_{n-1}\right)-k_{n-1}sgn\left(z_{n-1}\right))+\left({\tilde{x}}_{n-1}-z_{n-1}\right)y_{r}^{\left(n-1\right)}\\ & \leq & -\left(1-\zeta \right)\sum_{i=1}^{n-2}z_{i}^{2}+z_{n-1}z_{n}+z_{n-1}(\left(z_{n-2}-{\hat{z}}_{n-2}\right)-c_{n-1}z_{n-1}+c_{n-1}\left(z_{n-1}-{\hat{z}}_{n-1}\right)\\ \; & \; & +f_{n-1}\left({\underline{x}}_{n-1}\right)-f_{n-1}\left({\underline{\hat{x}}}_{n-1}\right))+\left({\tilde{x}}_{n-1}-z_{n-1}\right)y_{r}^{\left(n-1\right)}-k_{n-1}\left|z_{n-1}\right|\\ & \leq & -\left(1-\zeta \right)\sum_{i=1}^{n-2}z_{i}^{2}+z_{n-1}z_{n}-c_{n-1}z_{n-1}^{2}+z_{n-1}\left({\tilde{x}}_{n-2}+c_{n-1}{\tilde{x}}_{n-1}+L_{n-1}{\underline{\tilde{x}}}_{n-1}\right)\\ \; & \; & +\left({\tilde{x}}_{n-1}-z_{n-1}\right)y_{r}^{\left(n-1\right)}-k_{n-1}\left|z_{n-1}\right|\\ \; & \leq & -\left(1-\zeta \right)\sum_{i=1}^{n-2}z_{i}^{2}+z_{n-1}z_{n}-c_{n-1}z_{n-1}^{2}+|z_{n-1}|\left(1+c_{n-1}+L_{n-1}\right)\left|\right|{\underline{\tilde{x}}}_{n-1}\left|\right|\\ \; & \; & +\left|\right|{\underline{\tilde{x}}}_{n-1}\left||\right|y_{r}^{\left(n-1\right)}|-(k_{n-1}-|y_{r}^{\left(n-1\right)}\left|)\right|z_{n-1}| \end{array}$$If the event-triggered condition is designed as (29), (Equation 45) can be calculated as:
(46)$$\begin{array}{ccc} {\dot{V}}_{n-1} & \leq & -\left(1-\zeta \right)\sum_{i=1}^{n-2}z_{i}^{2}+z_{n-1}z_{n}-c_{n-1}z_{n-1}^{2}+c_{n-1}\zeta z_{n-1}^{2}\\ \; & \; & -(k_{n-1}-|y_{r}^{\left(n-1\right)}|+\frac{c_{n-1}}{1+c_{n-1}+l_{n-1}}\zeta |y_{r}^{\left(n-1\right)}\left|)\right|z_{n-1}| \end{array}$$Together with Assumption 2, that the boundedness of $y_{r}^{\left(i\right)},\;\;i=1,\cdots n-1$, we can design $k_{n-1}$ with $k_{n-1}-|y_{r}^{\left(n-1\right)}|+\frac{c_{n-1}}{1+c_{n-1}+l_{n-1}}\zeta \left|y_{r}^{\left(n-1\right)}\right|>0$, so that
(47)$$\begin{array}{ccc} {\dot{V}}_{n-1} & \leq & -\left(1-\zeta \right)\sum_{i=1}^{n-1}z_{i}^{2}+z_{n-1}z_{n} \end{array}$$Step n. In this step, we choose the Lyapunov function as:
(48)$$\begin{array}{c} V_{n}=V_{n-1}+\frac{1}{2}z_{n}^{2}+\frac{1}{2}{\hat{z}}_{n}^{2}+\frac{1}{2\beta }{\tilde{a}}^{2}+\frac{\lambda b}{2\gamma }{\tilde{h}}^{2}+\frac{1}{2\delta }{\tilde{D}}^{2} \end{array}$$Calculating the time derivative of *z* along the solutions of System (1), it follows from Assumptions 1 and 2 that:
(49)$$\begin{array}{ccc} z_{n}{\dot{z}}_{n} & = & z_{n}\left(\lambda bu-y_{r}^{\left(n\right)}+af_{n}+d\right)=z_{n}\left(af_{n}+d-y_{r}^{\left(n\right)}+\bar{u}-\lambda b\tilde{h}\bar{u}\right)\\ \; & = & z_{n}\left(af_{n}+d-y_{r}^{\left(n\right)}-\lambda b\tilde{h}\bar{u}-c_{n}{\hat{z}}_{n}-\hat{a}f_{n}\left({\hat{\underline{x}}}_{n}\right)-{\hat{z}}_{n-1}+y_{r}^{\left(n\right)}-sign\left(z_{n}\right)\hat{D}\right)\\ \; & = & z_{n}(-c_{n}{\hat{z}}_{n}-{\hat{z}}_{n-1}+af_{n}\left({\underline{x}}_{n}\right)-af_{n}\left({\hat{\underline{x}}}_{n}\right)+af_{n}\left({\hat{\underline{x}}}_{n}\right)-\hat{a}f_{n}\left({\hat{\underline{x}}}_{n}\right)\\ \; & \; & +d-sign\left(z_{n}\right)D+sign\left(z_{n}\right)\tilde{D}-\lambda b\tilde{h}\bar{u})\\ \; & \leq & z_{n}(-c_{n}z_{n}+c_{n}\left(z_{n}-{\hat{z}}_{n}\right)-z_{n-1}+\left(z_{n-1}-{\hat{z}}_{n-1}\right)+a\left(f_{n}\left({\underline{x}}_{n}\right)-f_{n}\left({\hat{\underline{x}}}_{n}\right)\right)\\ \; & \; & +\tilde{a}f_{n}\left({\underline{x}}_{n}\right)+sign\left(z_{n}\right)\tilde{D}-\lambda b\tilde{h}\bar{u})\\ \; & \leq & -c_{n}z_{n}^{2}-z_{n}z_{n-1}+|z_{n}|(c_{n}|{\tilde{x}}_{n}\left|+\right|{\tilde{x}}_{n-1}|+al_{n}\left|\right|{\tilde{\underline{x}}}_{n}\left||\right)+z_{n}\tilde{a}f_{n}\left({\underline{x}}_{n}\right)+z_{n}sign\left(z_{n}\right)\tilde{D}-z_{n}\lambda b\tilde{h}\bar{u}\\ \; & \leq & -c_{n}z_{n}^{2}-z_{n}z_{n-1}+|z_{n}|\left(1+c_{n}+al_{n}\right)\left|\right|{\tilde{\underline{x}}}_{n}\left|\right|+z_{n}\tilde{a}f_{n}\left({\underline{x}}_{n}\right)+z_{n}sign\left(z_{n}\right)\tilde{D}-z_{n}\lambda b\tilde{h}\bar{u} \end{array}$$If the event-triggered condition is designed as (29), (Equation 49) can be calculated as:
(50)$$\begin{array}{ccc} z_{n}{\dot{z}}_{n} & \leq & -c_{n}z_{n}^{2}-z_{n}z_{n-1}+\frac{1+c_{n}+al_{n}}{1+c_{n}+\bar{a}l_{n}}c_{n}\zeta z_{n}^{2}+z_{n}\tilde{a}f_{n}\left({\underline{x}}_{n}\right)+z_{n}sgn\left(z_{n}\right)\tilde{D}-z_{n}\lambda b\tilde{h}\bar{u}\\ \; & \leq & -\left(1-\zeta \right)c_{n}z_{n}^{2}-z_{n}z_{n-1}+z_{n}\tilde{a}f_{n}\left({\underline{x}}_{n}\right)+z_{n}sign\left(z_{n}\right)\tilde{D}-z_{n}\lambda b\tilde{h}\bar{u} \end{array}$$In addition, we can obtain:
(51)$$\begin{array}{c} {\hat{z}}_{n}{\dot{\hat{z}}}_{n}=-{\hat{z}}_{n}y_{r}^{\left(n\right)}=-\left(z_{n}-{\tilde{z}}_{n}\right)y_{r}^{\left(n\right)}=\left({\tilde{x}}_{n}-z_{n}\right)y_{r}^{\left(n\right)}\leq |z_{n}\left|\right|y_{r}^{\left(n\right)}|+\frac{c_{n}}{1+c_{n}+\bar{a}l_{n}}\zeta ∥z_{n}∥|y_{r}^{\left(n\right)}| \end{array}$$With (Equation 47), (Equation 50), and (Equation 51) in mind, calculating the time derivative of (Equation 48) along the proposed adaptive fault-tolerant controller (Equation 33) and the parameter update laws (Equation 20)–(22), we can obtain the following inequality:
(52)$$\begin{array}{ccc} {\dot{V}}_{n} & \leq & -\left(1-\zeta \right)\sum_{i=1}^{n}z_{i}^{2}+\tilde{a}\left(z_{n}f_{n}\left({\hat{\underline{x}}}_{n}\right)-\frac{1}{\beta }\dot{\hat{a}}\right)-\lambda b\tilde{h}\left(z_{n}\bar{u}+\frac{1}{\gamma }\dot{\hat{h}}\right)-(k_{n}-\left|y_{r}^{\left(n\right)}\right|\\ \; & \; & +\frac{c_{n}}{1+c_{n}+\bar{a}l_{n}}\zeta |y_{r}^{\left(n\right)}\left|)\right|z_{n}|+\tilde{D}\left(z_{n}sgn\left(z_{n}\right)-\frac{1}{\delta }\dot{\hat{D}}\right) \end{array}$$If we choose $k_{n}$ with $k_{n}-|y_{r}^{\left(n\right)}|+\frac{c_{n}}{1+c_{n}+\bar{a}l_{n}}\zeta \left|y_{r}^{\left(n\right)}\right|>0$,
(53)$$\begin{array}{ccc} {\dot{V}}_{n} & \leq & -\left(1-\zeta \right)\sum_{i=1}^{n}z_{i}^{2} \end{array}$$Case 2: At the event-sampled instants $t=t_{k},\;k\in Z^{+}$, we will verify that $V\left(Z\left(t_{k+1}\right)\right)<V\left(Z\left(t_{k}\right)\right)$. In fact, since V(Z(t)) is continuous and $\dot{V}\left(Z\left(t\right)\right)<0,$ for $t\in [t_{k},t_{k+1}),$ hence for any sequence $\left\{t_{ik}\right\}\subset \left[\frac{t_{k}+t_{k+1}}{2},t_{k+1}\right)$, we have $\dot{V}\left(Z\left(t_{ik}\right)\right)<0.$ We choose an increasing subsequence $\left\{t_{jik}\right\}\subset \left\{t_{i}k\right\}$ and $\lim_{j\to \infty }t_{jik}=t_{k+1}$. Then, it is obvious to prove that $\left\{t_{jik}\right\}\subset \left[\frac{t_{k}+t_{k+1}}{2},t_{k+1}\right)$ and $V\left(Z\left(t_{k}\right)\right)>V(Z\left(t_{jik}\right)$. Taking the limit of both sides of the above inequality, we can obtain:
(54)$$\begin{array}{c} V\left(Z\left(t_{k+1}\right)\right)=\lim_{j\to \infty }V\left(Z\left(t_{jik}\right)\right)<V\left(Z\left(t_{k}\right)\right) \end{array}$$By Case 1 (Equation 53) and Case 2 (Equation 54), it is easy to verify that ${\dot{V}}_{n}<0$ for ∀z≠0. Combining with (Equation 48), the closed-loop system signal $z_{i}$ is globally bounded. By invoking the LaSalle invariance principle, we can conclude that $\lim_{t\to \infty }z_{i}\left(t\right)=0,\;i=1,\cdots n$. It is shown that the asymptotic tracking is achieved. ☐

**Remark** **6.**
*From (Equation 33), the proposed adaptive fault-tolerant controller is constant at some inter-event intervals. This is the ideal result in the remote actual control. The network transmission resources are effectively saved, and it is robust for the packet loss problem. Then, the proposed control algorithm is also fault tolerant to actuator faults caused by electromagnetic interference in network transmission. Furthermore, different from the traditional backstepping controller (Equation 15), the event-triggered controller (Equation 33) can overcome the computational explosion caused by the higher order of the virtual control law $\alpha_{i}$.*


**Remark** **7.**
*Inspired by [38], when the dead-zone input is considered in the actuator channel:*
(55)$$\begin{array}{c} u_{f}\left(t\right)=\left\{\begin{array}{c} m_{r}\left(u\left(t\right)-b_{r}\right),\;\;\;\;u\left(t\right)\geq b_{r}\\ 0\;\;\;\;\;\;\;\;\;\;\;\;\;\;\;\;-b_{l}<u\left(t\right)<b_{r}\\ m_{l}\left(u\left(t\right)+b_{l}\right),\;\;\;\;u\left(t\right)\leq -b_{l} \end{array}\right. \end{array}$$
*where the interval $\left[-b_{l},\;b_{r}\right]$ is the dead-zone. Defining $\lambda =m_{r}=m_{l}>0,\;b_{r},\;b_{l}>0$, the dead-zone fault model can be described as:*
(56)$$\begin{array}{c} u_{f}\left(t\right)=\lambda u\left(t\right)+\mu \left(t\right) \end{array}$$
*where:*
(57)$$\begin{array}{c} \mu \left(t\right)=\left\{\begin{array}{c} -m_{r}b_{r},\;\;\;\;u\left(t\right)\geq b_{r}\\ -\lambda u\left(t\right)\;\;\;\;-b_{l}<u\left(t\right)<b_{r}\\ m_{l}b_{l},\;\;\;\;u\left(t\right)\leq -b_{l} \end{array}\right. \end{array}$$

*Then, it is consistent with the fault model (Equation 2). Therefore, The main result of this paper can be extended to solve the problem of dead-zone failures.*


**Remark** **8.**
*It is clear in Figure 1 that the parameter update laws (Equation 20)–(22) produced system sensor. The real-time update transmission of parameter signals may cause network congestion, and we can solve the problem with the following method.*

*The adaptive update laws are redesigned as:*
(58)$$\begin{array}{c} \dot{\hat{a}}=\beta {\hat{z}}_{n}f_{n}\left({\hat{\underline{x}}}_{n}\right) \end{array}$$
(59)$$\begin{array}{c} \dot{\hat{h}}=-\gamma {\hat{z}}_{n}\bar{u} \end{array}$$
(60)$$\begin{array}{c} \dot{\hat{D}}=\delta {\hat{z}}_{n}sgn\left(z_{n}\right) \end{array}$$

*The parameter estimation error satisfies:*
(61)$$\begin{array}{ccc} \tilde{a}\left(z_{n}f_{n}\left({\hat{\underline{x}}}_{n}\right)-\frac{1}{\beta }\dot{\hat{a}}\right) & = & \tilde{a}\left(z_{n}f_{n}\left({\hat{\underline{x}}}_{n}\right)-{\hat{z}}_{n}f_{n}\left({\hat{\underline{x}}}_{n}\right)\right)=\tilde{a}f_{n}\left({\hat{\underline{x}}}_{n}\right)\left(z_{n}-{\hat{z}}_{n}\right)=\tilde{a}f_{n}\left({\hat{\underline{x}}}_{n}\right){\tilde{z}}_{n}=\tilde{a}f_{n}\left({\hat{\underline{x}}}_{n}\right){\tilde{x}}_{n}\\ \; & \leq & |\tilde{a}f_{n}\left({\hat{\underline{x}}}_{n}\right)\left|\right|{\tilde{x}}_{n}\left|\leq p\right|\tilde{a}f_{n}\left({\hat{\underline{x}}}_{n}\right)\left|\right|z_{n}|\leq p_{max}\left|z_{n}\right| \end{array}$$
*where p and $p_{max}$ are selected to ensure that the above inequality holds. The new event-triggered condition guarantees that the parameter exists.*

*The following equation can be calculated in a similar manner:*
(62)$$\begin{array}{c} \lambda b\tilde{h}\left(z_{n}\bar{u}+\frac{1}{\gamma }\dot{\hat{h}}\right)=\lambda b\tilde{h}\bar{u}\left(z_{n}-{\hat{z}}_{n}\right)\leq |\lambda b\tilde{h}\bar{u}\left|p\right|z_{n}|\leq p_{max}\left|z_{n}\right| \end{array}$$
(63)$$\begin{array}{c} \tilde{D}\left(z_{n}sgn\left(z_{n}\right)-\frac{1}{\delta }\dot{\hat{D}}\right)=\tilde{D}sgn\left(z_{n}\right)\left(z_{n}-{\hat{z}}_{n}\right)\leq |\tilde{D}\left|p\right|z_{n}|\leq p_{max}\left|z_{n}\right| \end{array}$$

*These three inequalities can be applied directly to the proof of Theorem 1. Hence, we can realize the event-triggered adaptive parameter update law. The basic architecture for the improved event-triggered adaptive fault-tolerant control algorithm is illustrated by Figure 2.*


Next, we will show that the proposed control strategy can avoid the Zeno behavior. $t_{k+1}-t_{k}>T$:(64)$$\begin{array}{c} |f_{i}\left({\underline{x}}_{i}\right)-f_{i}\left({\hat{\underline{x}}}_{i}\right)|\leq L_{i}∥{\tilde{\underline{x}}}_{i}∥\leq \frac{l_{i}c_{i}\zeta }{1+c_{i}+l_{i}}\left|z_{i}\right|<{\bar{z}}_{i} \end{array}$$

It is easy to know that $f_{i}\left({\hat{\underline{x}}}_{i}\right)$ are bounded and $|f_{i}\left({\underline{x}}_{i}\right)|<F_{i}$.
(65)$$\begin{array}{c} \frac{d}{dt}∥x∥=\frac{d}{dt}{\left(x^{T}x\right)}^{\frac{1}{2}}=\frac{x^{T}\dot{x}}{∥x∥}\leq ∥\dot{x}∥ \end{array}$$

Considering the event-triggered error dynamics:(66)$$\begin{array}{ccc} {\dot{\tilde{x}}}_{i} & = & {\dot{x}}_{i}\left(t\right)-{\dot{\hat{x}}}_{i}\left(t\right)=x_{i+1}+f_{i}\left({\underline{x}}_{i}\right) \end{array}$$(67)$$\begin{array}{ccc} \frac{d}{dt}∥\tilde{x}∥\leq ∥\dot{\tilde{x}}\left|\right| & = & ∥A\tilde{x}+f∥\leq ∥A∥∥\tilde{x}∥+F \end{array}$$
(68)$$\begin{array}{c} ∥\tilde{x}∥\leq \int_{t_{k}}^{t}e^{∥A∥(t-\tau )}∥F∥dt=\frac{∥F∥}{∥A∥}\left(e^{∥A∥(t-t_{k})}-1\right),\;\;\;\;t\in \left[t_{k},\;t_{k+1}\right) \end{array}$$
it can be calculated that:(69)$$\begin{array}{c} T=t_{k+1}-t_{k}\geq t-t_{k}=\frac{1}{∥A∥}\ln(\frac{∥A∥}{∥F∥}(∥\tilde{x}∥+1))\geq 0\;\;\;\; \end{array}$$

Thus, the inter-event times $t_{k+1}-t_{k}$ are lower bounded by a nonzero positive constant *T*, that is the Zeno behavior does not occur.

## 4. Simulation

In this section, a fan speed control system is used to illustrate the effectiveness of the designed event-triggered adaptive fault-tolerant controller. The fan speed control system presented in reference [32] is subject to external disturbance and actuator faults. It is shown that even with unknown faults in the armature voltage, the proposed speed event-triggered adaptive fault-tolerant controller can realize the asymptotic regulation control of the desired reference signal for the fan speed.

The dynamics of a fan driven by a DC motor is described by:(70)$$\begin{array}{c} J_{1}\dot{v}=k_{1}I-\tau_{L}-\tau_{D}\left(v\right)\\ J_{2}\dot{I}=u-k_{2}v-RI+d_{e}\\ y=v \end{array}$$
where *v* is the fan speed, *I* is the armature current, $\tau_{L}$ is the constant load torque, $\tau_{D}\left(v\right)$ is the drag torque, *u* is the armature voltage, which is considered as the input, $J_{1},k_{1},k_{2},R$ are known positive constants, and the inductance $J_{2}$ may be an unknown constant. The function $d_{e}$ represents the uncertain external disturbance. The control task is the set-point regulation control of the fan speed *v*, against the unknown faults and the disturbance $d_{e}$.

In order to realize the control objective, we introduce the change of coordinates:(71)$$\begin{array}{c} x_{1}=v,\;\;x_{2}=\frac{k_{1}}{J_{1}}I \end{array}$$

With the new coordinates, the fan speed control system (Equation 70) is turned into:(72)$$\begin{array}{ccc} {\dot{x}}_{1} & = & x_{2}-\frac{1}{J_{1}}\tau_{0}\left(x_{1}\right)-\frac{1}{J_{1}}\tau_{L}\\ {\dot{x}}_{2} & = & -\frac{k_{1}k_{2}}{J_{1}J_{2}}x_{1}-\frac{R}{J_{2}}x_{2}+\frac{k_{1}}{J_{1}J_{2}}u+\frac{1}{J_{2}}d_{e}\\ y & = & x_{1}. \end{array}$$

Clearly, System (Equation 72) is the strict feedback form in System (1) with:(73)$$\begin{array}{c} f_{1}\left({\underline{x}}_{1}\right)=\frac{1}{J_{1}}\tau_{0}\left(x_{1}\right)-\frac{1}{J_{1}}\tau_{L},\;\;a=\frac{1}{J_{2}},\;\;f_{2}\left(x\right)=-\frac{k_{1}k_{2}}{J_{1}}x_{1}-Rx_{2},\;\;b=\frac{k_{1}}{J_{1}J_{2}},\;\;d_{0}=\frac{1}{J_{2}}d_{e} \end{array}$$

We set the system model system as $\tau_{0}\left(x_{1}\right)=\sin\left(x_{1}\right),\;\;\tau_{L}=J_{1}=k_{1}=k_{2}=R=1$, the initial conditions $x_{1}\left(0\right)=-9.6,\;\;x_{2}\left(0\right)=0.6$, and the external disturbance $d_{e}=10$. We choose the healthy indicator λ=0.8 and the bias fault μ=2. Under the event-triggered adaptive fault-tolerant control law (33) with the events triggering condition (27)–(29) and the traditional continuous time fault-tolerant control law (15)–(16) with the parameter update laws (Equation 20)–(22), the simulation comparison results are described in Figure 3, Figure 4 and Figure 5.

From the simulation results shown in Figure 3 and Figure 4, it can be seen that all the states are globally bounded and convergent, then the good set-point regulation performance can be achieved by the proposed methodology in the paper. In this way, the regulation of the fan speed *v* is achieved, even under the external disturbance in the armature voltage and the actuator faults. Moreover, Figure 5 shows the trajectories of the control signals u(t) under event-triggered control and continuous time control. The event-triggered control signal are constant in some time intervals, then the proposed event-triggered FTC mechanism can increase the utilization of valuable network resources. Therefore, an appropriate compromise is realized between control performance and real-time transmission consumption.

## 5. Conclusions

This paper addressed the problem of the asymptotic tracking by using event-triggered adaptive fault-tolerant control for uncertain systems with actuator failures and external disturbance. The proposed system model was general enough to express many extended chained form systems and practical nonlinear examples. By using the the event-triggered theory, the adaptive backstepping technique, and Lyapunov stability theory, this paper presented a novel fault-tolerant control strategy. It was shown that the proposed control laws guaranteed an appropriate comprise between the control performance and the sensor data real-time transmission consumption. Meanwhile, the convergence could be realized even with the actuator faults by the simple fault-tolerant control law. One of the future research topics would be the extension of the presented method to more general cases; for example, the finite-time FTC for the systems subject to the actuator saturation, small delays, and sampling noise.

## Figures and Tables

**Figure 1 entropy-22-00598-f001:**
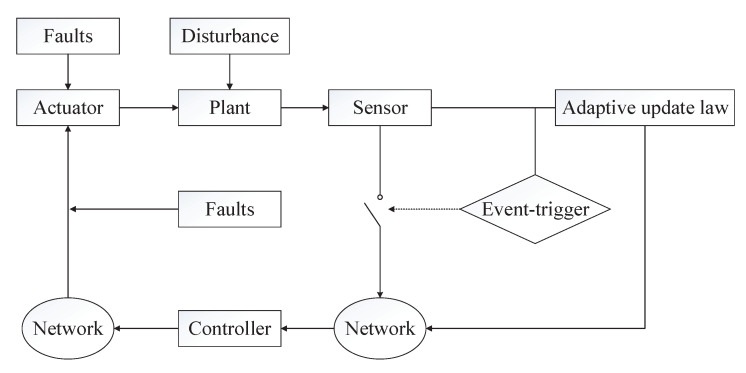
Architecture for event-triggered adaptive fault-tolerant control.

**Figure 2 entropy-22-00598-f002:**
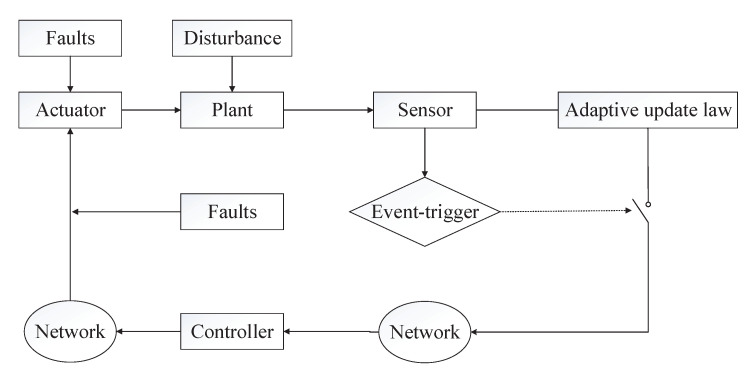
Architecture for the improved event-triggered adaptive fault-tolerant control.

**Figure 3 entropy-22-00598-f003:**
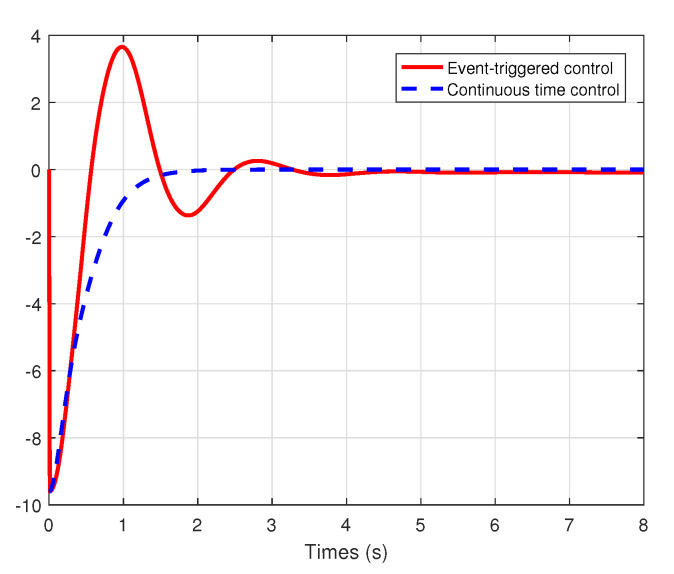
The trajectories of the states $x_{1}\left(t\right)$ under event-triggered control and continuous time control.

**Figure 4 entropy-22-00598-f004:**
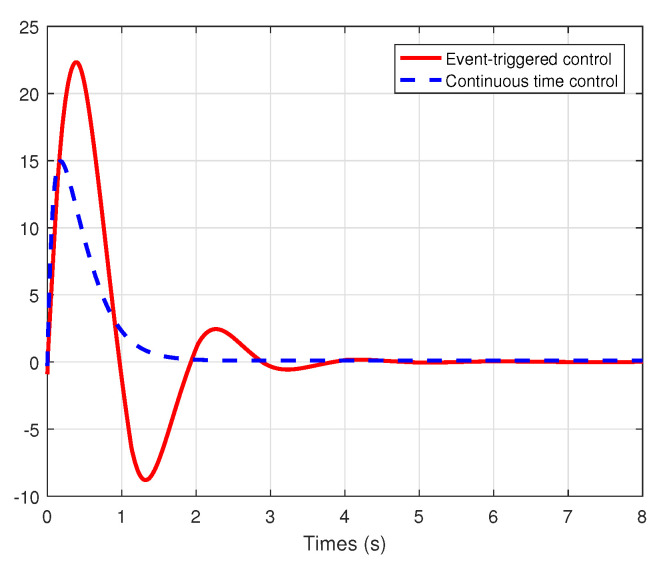
The trajectories of the states $x_{2}\left(t\right)$ under event-triggered control and continuous time control.

**Figure 5 entropy-22-00598-f005:**
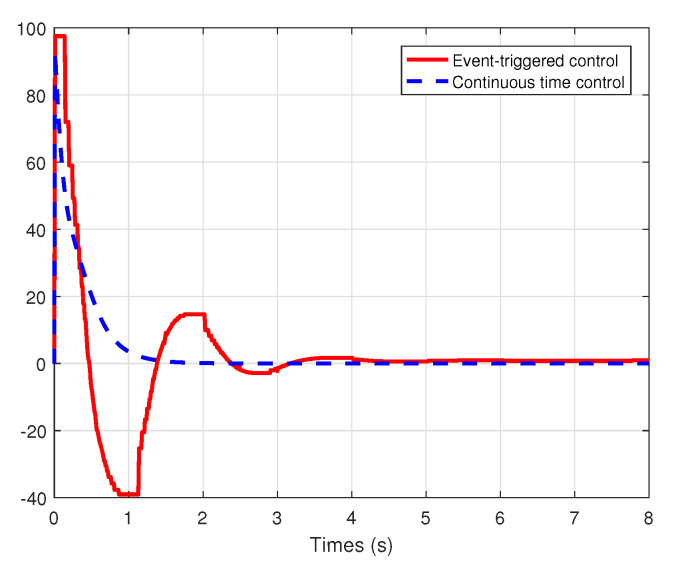
u(t) under event-triggered control and continuous time control.

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
