# Peer review of "Event-Triggered Adaptive Fault Tolerant Control for a Class of Uncertain Nonlinear Systems"

_entropy, 2020, doi:10.3390/e22060598_

Round 1

Reviewer 1 Report

Comments to the authors

Manuscript number: entropy-801053

Title: Event-triggered adaptive fault tolerant control for a class of uncertain nonlinear systems

1) English issues:

a) … eliminating the impact [of] the failures.

b) … the above mention[ed] event-triggered methods …

c) … are [is] assumed to bounded, that is …

2) Regarding the application of FTC scheme, there are many works in the literature that can be discussed by the authors to improve the literature review. For example, the application of FTC schemes in microgrids.

a) “Fault-tolerant supervisory controller for a hybrid ac/dc micro-grid,” IEEE Transactions on Smart Grid, 2018.

b) “Fault tolerant predictive control design for reliable microgrid energy management under uncertainties,” Energy, 2015.

3) Since uf is the control input in the presence of faults, the explanation after Equation (1) should be “y is the system output, and uf  is the control input under faulty condition.”

4) As “y” is used to denote the output of the system, avoid using it in Assumption 1 to refer to the states of the system. Please use another notation in Assumption 1.

5)  Section 2.2 is the basic idea of backstepping algorithm that can be found in any control book. The reviewer does not see the necessity of discussing it in the manuscript.

6) Simulation section should be improved. The effectiveness of the methods has not been discussed. The performance of the nominal controller in the presence of faults should be compared with that of the proposed fault tolerant scheme.

7) The authors are recommended to briefly discuss pros and cons of the proposed method in the conclusion.

Reviewer 2 Report

This paper presents an event-triggered adaptive fault tolerant control for uncertain strict-feedback nonlinear systems.

1. Using event-triggered laws, how are (38), (41), and (46) derived ? These procedures are questionable.
The reasonable explanations are required in detail.

2. l_{i} should be L_{i} in (27)--(29) and the corresponding equations. Notations should be checked through all pages of this paper.

3. Employing sgn functions in the adaptive laws is a general result for the asymptotical convergence.
However, the use of sgn functions has the chattering problem in the control input.

4. The paper has some typos and language issue which needs to be checked and corrected in the revision. English needs to be polished further.

Round 2

Reviewer 1 Report

The manuscript is acceptable in the present form. 

Reviewer 2 Report

The event triggering errors satisfy \tilde{x}_{i}(t_{k}) = 0 for k \in Z^{+} and the triggering errors \tilde{x}_{i} should be less than its threshold. Thus, the inequalities of the event triggereing conditions (27)--(29) should be reversed (i.e., \|\tilde{x}_{i}\| >= ... ).